# The Relationship between the Contouring Time of the Metal Artifacts Area and Metal Artifacts in Head and Neck Radiotherapy

**Kouji Katsura** [1,2,*], **Satoshi Tanabe** [3], **Hisashi Nakano** [3], **Madoka Sakai** [4], **Atsushi Ohta** [4], **Motoki Kaidu** [4], **Marie Soga** [1], **Taichi Kobayashi** [2], **Masaki Takamura** [2] **and Takafumi Hayashi** [1,2]

1 Department of Oral Radiology, Niigata University Medical and Dental Hospital, Niigata 951-8520, Japan
2 Division of Oral and Maxillofacial Radiology, Faculty of Dentistry & Graduate School of Medical and Dental Sciences, Niigata University, Niigata 951-8514, Japan
3 Division of Radiation Oncology, Niigata University Medical and Dental Hospital, Niigata 851-8520, Japan
4 Department of Radiology and Radiation Oncology, Graduate School of Medical and Dental Sciences, Niigata University, Niigata 951-8514, Japan
* Correspondence: katsu@dent.niigata-u.ac.jp; Tel.: +81-25-227-2914

**Abstract:** (1) Background: The impacts of metal artifacts (MAs) on the contouring workload for head and neck radiotherapy have not yet been clarified. Therefore, this study evaluated the relationship between the contouring time of the MAs area and MAs on head and neck radiotherapy treatment planning. (2) Methods: We used treatment planning computed tomography (CT) images for head and neck radiotherapy. MAs were classified into three severities by the percentage of CT images containing MAs: mild (<25%), moderate (25–75%), and severe (>75%). We randomly selected nine patients to evaluate the relationship between MAs and the contouring time of the MAs area. (3) Results: The contouring time of MAs showed moderate positive correlations with the MAs volume and the number of CT images containing MAs. Interobserver reliability of the extracted MAs volume and contouring time were excellent and poor, respectively. (4) Conclusions: Our study suggests that the contouring time of MAs areas is related to individual commitment rather than clinical experience. Therefore, the development of software combining metal artifact reduction methods with automatic contouring methods is necessary to reducing interobserver variability and contouring workload.

**Keywords:** metal artifacts; metallic dental restorations; contouring workload; head and neck radiotherapy





## 1. Introduction

Radiotherapy is a widely used treatment modality for head and neck cancer. However, metallic dental restorations (MDRs) make it difficult to contour the target volumes (TVs) and organs at risk (OARs) because MDRs generate metal artifacts (MAs) in radiotherapy treatment planning computed tomography (CT) images [1,2]. In addition, MAs decrease the accuracy of dose calculation [2–4].

Currently, there are two measures against MDRs for radiotherapy treatment planning [5]. One measure against MDRs is MDR removal, in which all MDRs are removed before radiotherapy treatment planning CT. Another is non-MDR removal, in which MAs are replaced with the CT value of soft tissue or water as much as possible and magnetic resonance imaging or positron emission tomography–CT is used to identify the TVs and OARs. In practice, MDR removal is often unrealistic for the following reasons: it is likely to cause tooth extraction, patients often refuse to undergo MDR removal, and no dental service is available within or around the cancer treatment facilities. In addition, the medical cost of MDR removal has been estimated to be higher than that of non-MDR removal [6]. On the other hand, non-MDR removal increases the contouring workload more than MDR removal. However, the impacts of MAs on the contouring workload for head and neck

radiotherapy have not yet been clarified. Therefore, this study evaluated the relationship between the contouring time of the MAs area and MAs on head and neck radiotherapy treatment planning.

## 2. Materials and Methods

### 2.1. Patients

Patients who underwent two-step IMRT (70 Gy in 35 fractions) for head and neck cancer at our institution between January 2018 and December 2020 were enrolled in this study. Patients without MAs in the oral cavity were excluded. In this study, CT slices that included the oral cavity were defined as the area between the root apex of the maxillary and mandibular canines. Streaking artifacts of less than $-200$ and more than 400 Hounsfield units (HU) generated from MDRs were defined as MAs. MAs in the oral cavity were classified by an oral radiologist (K.K.) into three severities by the percentage of CT images containing MAs: mild (<25%), moderate (25–75%), and severe (>75%). We randomly selected a total of nine patients, with three patients from each severity group. The calculation formula was as follows:

(i) The percentage of CT images containing MAs in the oral cavity

$$= 100 \times \frac{the\ number\ of\ CT\ slices\ containing\ MAs\ in\ the\ oral\ cavity}{the\ number\ of\ CT\ slices\ in\ the\ oral\ cavity}$$

Informed consent was obtained as an opt-out approach on our institutional website, on which information about the study objectives and procedures was published, instead of using written informed consent. This study was approved by the institutional review board of our hospital (no. 202-0265).

### 2.2. Radiotherapy Treatment Planning Computed Tomography Data

Each patient was immobilized using a thermoplastic mask (CIVCO Co., Orange City, IA, USA) and a mouthpiece composed of an ethylene–vinyl acetate copolymer (Erkodent Erich Kopp GmbH, Pfalzgrafenweiler, Germany). Then, a radiotherapy treatment planning CT scan was conducted using a 16-slice CT scanner (Lightspeed RT, General Electric Medical Systems, Waukesha, WI, USA) with the following parameters: tube voltage, 120 kVp; tube current, auto-exposure control; slice thickness, 1.25 mm; and field of view, 50 cm. All CT image data were reconstructed with a thickness of 2.5 mm.

### 2.3. Contouring of the Dental Metal Artifacts Area

Radiotherapy treatment planning was conducted using an Eclipse treatment planning system (version 15.5; Varian Medical Systems Inc., Palo Alto, CA, USA). The process of contouring works, including measures against MAs, is shown in Figure 1. Based on the radiotherapy treatment planning CT images (Figure 1a), three medical physicists (clinical experience: 1, 4, and 12 years, respectively) contoured the MAs area. Streaking artifacts of less than $-200$ and more than 400 Hounsfield units (HU) generated from MDRs as the MAs area were manually extracted as much as possible. Finally, the CT values of the extracted MAs areas were replaced with a value of 0 HU (Figure 1b).

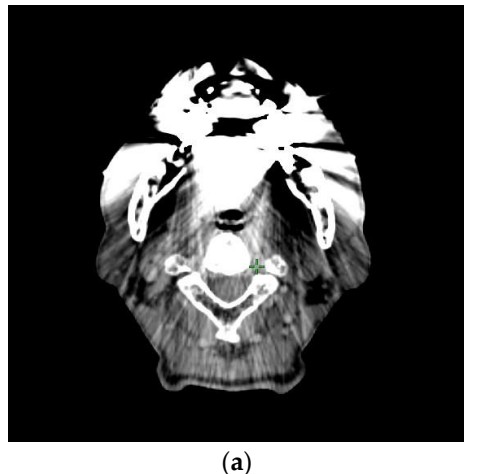 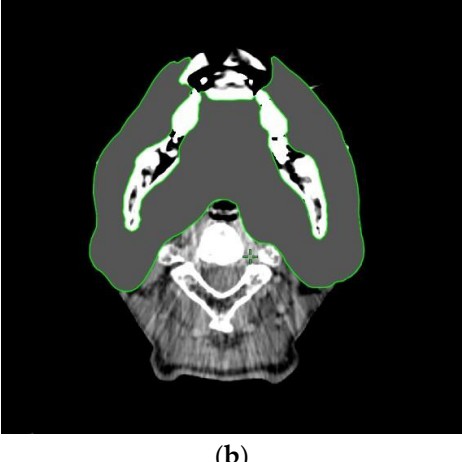

(**a**) (**b**)

**Figure 1.** The process of contouring works of metal artifacts (MAs) for head and neck radiotherapy. (**a**) Radiotherapy treatment planning computed tomography (CT) images with MAs. (**b**) Replacement of MAs with Hounsfield units equal to water.

*2.4. Evaluation*

We defined the contouring time as the time from when a contour was selected to edit the artifact to when the contour-editing window in Eclipse was closed. The relationships between the severities of MAs and the volume of extracted MAs, as well as the severities of MAs and the contouring time of MAs, were evaluated using the Kruskal–Wallis test and Bonferroni's post hoc test. In addition, the correlations with the contouring time of MAs to the volume of extracted MAs and the number of images containing MAs were evaluated using Spearman's rank correlation coefficient. Correlation coefficient (r) values were interpreted as $r \geq 0.7$, $0.7 > r > 0.4$, and $r \leq 0.4$ as strong, moderate, and weak, respectively. The interobserver reliability of the contouring for the MAs area was evaluated using the intraclass correlation coefficient (ICC) (Model 2,1). The ICC values were interpreted as >0.90, 0.9.0–0.75, 0.75–0.50, and <0.50 as excellent, good, fair, and poor, respectively [7]. Statistical analyses were conducted using IBM SPSS Statistics 23.0 for Windows (IBM Japan Ltd., Tokyo, Japan). *p*-values < 0.05 were considered significant.

**3. Results**

The characteristics of the selected patients are shown in Table 1. The median number of CT images in the oral cavity was 24 (range: 21–27). In addition, the median number of CT images containing MAs, the median volume of extracted MAs, and the median contouring time of the MAs areas were 11 images (range: 4–18), 105.3 cc (range: 16.8–245.4), and 19.3 min (range: 6.4–72.7), respectively.

**Table 1.** Characteristics of the selected patients.

| Characteristics | N = 9 |
| --- | --- |
| Age, range (median) | 55–76 years (66) |
| Gender | |
|     Male | 9 |
| Female | 0 |
| Primary tumor site | |
|     Nasopharynx | 2 |
| Oropharynx | 4 |
| Larynx | 2 |
| Maxillary sinus | 1 |
| Tumor classification | |
|     T1 | 3 |
|     T2 | 2 |
|     T3 | 2 |
|     T4a | 1 |
|     T4b | 1 |

**Table 1.** *Cont.*

| Characteristics | N = 9 |
|---|---|
| Node classification | |
| N0 | 1 |
| N1 | 4 |
| N2b | 3 |
| N2c | 1 |
| Cancer stage | |
| I | 1 |
| II | 2 |
| III | 2 |
| IVA | 3 |
| IVB | 1 |
| Number of CT images | |
| Oral cavity, range (median) | 21–26 images (24) |
| MAs, range (median) | 4–18 images (11) |
| Volume of extracted MAs, range (median) | 16.8–245.4 cc (105.3) |
| Contouring time of MAs area (median) | 6.4–72.7 min (19.3) |

CT = computed tomography, MAs = metal artifacts. The tumor and node classification and cancer stage followed the 8th edition of the UICC TNM classification and AJCC cancer staging system.

The relationships between the severities of MAs and the contouring time of the MAs areas, as well as the severities of MAs and the volume of extracted MAs, are shown in Table 2 and Figure 2. The volume of extracted MAs and the contouring time of the MAs area increased statistically significantly as the severities of MAs increased ($p = 0.001$ and $p = 0.014$, respectively).

**Table 2.** Relationships between the severities of MAs and the contouring time of the MAs area, and the severities of MAs and the volume of extracted MAs.

| | Severities of MAs | | | |
|---|---|---|---|---|
| | Mild (<25%) Median (Range) | Moderate (25–75%) Median (Range) | Severe (>75%) Median (Range) | *p* |
| Volume of extracted MAs | 49.2 cc (16.8–72.4) | 108.8 cc (53.9–142.7) | 209.8 cc (98.3–245.4) | <0.001 |
| Contouring time of MAs area | 16.1 min. (6.4–32.7) | 25.8 min. (11.4–46.2) | 40.3 min. (14.7–72.7) | <0.014 |

CT = computed tomography, MAs = metal artifacts, min. = minutes. The Kruskal–Wallis test was used for the analyses.

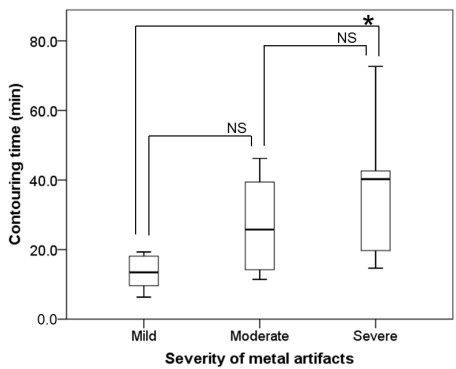

**Figure 2.** The relationship between the severities of metal artifacts (MAs) and the contouring time of the MAs area. A statistically significant difference between mild and severe MAs is shown ($p = 0.020$). Box plots indicate median, quartiles (25th and 75th percentiles), and extreme values for the contouring times in the three groups classified by the percentage of CT slices containing MAs. Bonferroni's post hoc test was used for the statistical analyses. The level of statistical significance was set at 0.05. * $p < 0.05$. NS = not significant.

The correlations between the contouring time of MAs to the volume of extracted MAs and the number of images containing MAs are shown in Table 3. The contouring times showed moderate positive correlations with the volume of extracted MAs ($r = 0.535$, $p = 0.004$) and the number of CT images containing MAs ($r = 0.596$, $p < 0.001$).

**Table 3.** Correlations with the contouring time of the MAs area to the volume of extracted MAs and the number of slices containing MAs.

|  | Contouring Time of MAs Area | *p* |
|---|---|---|
| Volume of extracted MAs | $r = 0.535$ | 0.004 |
| Number of CT slices containing MAs | $r = 0.596$ | <0.001 |

CT = computed tomography, MAs = metal artifacts. Spearman's rank correlation was used for the analyses.

The interobserver reliability of the volume of extracted MAs and the contouring time of the MAs area are shown in Table 4. The interobserver reliability was excellent (ICC = 0.934: 95% confidence interval = 0.78–0.98) for the volume of extracted MAs. On the other hand, the interobserver reliability was poor (ICC = 0.312: 95% confidence interval = −0.01–0.72) for the contouring times of the MAs area.

**Table 4.** Interobserver reliability of the extracted MAs volume and the MAs area contouring time.

| Observer (Clinical Experience) | Volume of Extracted MAs | | | Contouring Times of MAs Area | | |
|---|---|---|---|---|---|---|
|  | Volume Median (Range) | ICC | 95% CI | Contouring Time Median (Range) | ICC | 95% CI |
| A (1 year) B (4 years) C (12 years) | 96.9 cc (19.2–245.4) 112.9 cc (22.9–243.6) 127.6 cc (31.2–283.5) | 0.934 | 0.78–0.98 | 40.4 min. (16.1–53.2) 14.1 min. (6.4–25.4) 18.5 min. (9.1–90.9) | 0.312 | −0.01–0.72 |

MAs = metal artifacts, ICC = intraclass correlation coefficient, CI = confidence interval, min. = minutes. The intraclass correlation coefficient (Model 2,1) was used for the interobserver reliability.

## 4. Discussion

Accurate contouring of TVs and OARs is critical in head and neck radiotherapy because the contouring of these regions is directly associated with cancer control and the incidence and severity of radiation-induced adverse events. Therefore, the recent contouring of head and neck radiotherapy requires extensive time and expertise because of the number of OARs and complex anatomical structures and the measures against MAs. In fact, the German Society of Radiation Oncology reported that contouring was the most time-consuming procedure for head and neck radiotherapy [8]. Three previous studies evaluating the workload of radiotherapy treatment planning for head and neck radiotherapy reported that mean contouring times were 105 [8], 108 [9], and 180 min per patient [10]. Our study showed that the median contouring time of the MAs area was 19.3 min per patient. This contouring time of the MAs area is estimated to range from approximately 11% to 18% of the contouring time for head and neck radiotherapy treatment planning. In addition, a previous study reported that head and neck radiotherapy patients had dental restorations on approximately half of the teeth in the radiation field [11]. Therefore, the actual contouring time of the MAs area might be significantly longer than our results.

Our results showed that the contouring time of the MAs area and the volume of extracted MAs increased as the percentage of CT images containing MAs increased. Moreover, the contouring time of the MAs area showed moderate positive correlations with both the percentage of CT images containing MAs and the volume of extracted MAs. These results indicate that MAs have a negative impact on contouring workload for head and neck radiotherapy treatment planning. Therefore, measures against MAs are necessary for head and neck radiotherapy treatment. One strategy against MAs is MDR removal before radiotherapy treatment planning CT. However, it may not be possible to remove MDRs because of the possibility of tooth extraction, a lack of patient consent, or the unavailability of dental services in or around the cancer treatment facilities. Another strategy is to reduce MAs without MDRs using megavoltage CT (MVCT), dual energy computed tomography (DECT), or metal artifact reduction (MAR) methods.

MVCT can both improve the delineation of TVs and OARs in the MAs region and enable more accurate dose calculation for head and neck radiotherapy treatment planning [12]. However, MVCT has limitations such as poor soft tissue contrast, wide slice

thickness of radiotherapy planning CT scans, and increased doses received by the patient. DECT is a virtual mono-energetic image reconstruction technique using a high-energy and a low-energy X-ray. Some authors have reported that DECT could both reduce MAs [13] and positively affect the contouring workload of radiotherapy treatment planning [14]. However, they also reported that DECT could not remove strong MAs [13], and the dose in the MAs region calculated using DECT was not significantly improved versus the reference [14]. The MAR method detects the MAs area and replaces it with the estimated corrected HU value automatically. It has been reported that the MAR method both reduces the MAs volume and improves the accuracy of the HU of the MAs area [15]. In addition, the MAR method has been reported to improve the dose-calculation accuracy [16], alleviate uncertain delineation attributable to MAs [17], and improve the ease of contouring [18]. Therefore, the MAR method might be the best of these measures against MAs in head and neck radiotherapy treatment planning.

Our study only defined the HU of the MAs. Nevertheless, the volume of removed MAs had excellent reliability among the observers. On the other hand, regardless of clinical experience, the contouring time for measures against MAs had poor reliability among the observers. These results suggest that the manual contouring time of the MAs area is related to individual commitment rather than clinical experience. Several studies reported that automatic contouring could reduce the contouring time compared to manual contouring [9,10,19]. Van Dijk et al. reported that automatic contouring using deep-learning results was within or near the interobserver variability for manually edited contours [19], and Teguh et al. reported that an expert panel scored all automatic contouring as a "minor deviation, editable" or "better" compared to manual contouring [10]. Therefore, developing software combining the MAR method with the automatic contouring method is necessary to reduce the contouring workload and the interobserver variability in head and radiotherapy treatment planning.

Concerning the study limitations, this study only evaluated the relationship between the contouring time of the MAs area and MAs on the contouring workload for head and neck radiotherapy. Therefore, it is necessary to prospectively evaluate the impact of MAs on actual overall contouring time, including TVs and OARs' delineation in head and neck radiotherapy treatment planning.

## 5. Conclusions

Our study suggested that MAs have a large impact on the contouring times for head and neck radiotherapy treatment planning, and the manual contouring time of the MAs area is related to individual commitment rather than clinical experience. Therefore, it is necessary to evaluate the actual clinical impacts of MAs for head and neck radiotherapy treatment planning. In addition, we believe that it will be essential to develop software combining the MAR method with automatic contouring to reduce the contouring workload and interobserver variability for head and neck radiotherapy treatment planning.

**Author Contributions:** Conceptualization, K.K., S.T., H.N. and T.H.; methodology, K.K., S.T. and H.N.; validation, K.K., S.T. and H.N.; formal analysis, K.K.; investigation, K.K., S.T., H.N., M.S. (Madoka Sakai) and A.O.; resources, K.K., A.O., M.K., M.S. (Marie Soga), T.K. and M.T.; data curation, K.K., S.T. and T.H.; writing—original draft preparation, K.K.; writing—review and editing, K.K. and S.T.; visualization, K.K.; supervision, K.K., S.T., M.K. and T.H.; project administration, K.K. and S.T.; funding acquisition, K.K. and T.H. All authors have read and agreed to the published version of the manuscript.

**Funding:** This research received no external funding.

**Institutional Review Board Statement:** This study was conducted in accordance with the Declaration of Helsinki and approved by the Ethics Committee of Niigata University Medical and Dental Hospital (approval number 202-0265).

**Informed Consent Statement:** Informed consent was obtained in the form of an opt-out approach on our institutional website, on which information about the study objectives and procedures was published, instead of using written informed consent.

**Data Availability Statement:** Not applicable.

**Conflicts of Interest:** The authors declare no conflict of interest.

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
