# Peer review of "The Relationship between the Contouring Time of the Metal Artifacts Area and Metal Artifacts in Head and Neck Radiotherapy"

_tomography, doi:10.3390/tomography9010009_

Round 1

Reviewer 1 Report

1. General comments

The focus on how dental metal artifacts affect contouring in radiotherapy planning and the actual measurement and demonstration of contouring times is of great interest to the reader, especially to radiation therapists and medical physicists. The correlation between the degree and the contouring time is reasonable. As for the years of contour experience and contour time, the large individual differences were well understood from the actual situation in the field. First, as the author says, one way to reduce overall time spent on contouring to introduce an automatic contouring system. However, even an automatic contouring system is not perfect, and modifications are often necessary. In such cases, the degree of commitment will also have an impact, and it would be good to clarify the factors involved in commitment in future studies.

2. Specific comments

a) Major

None

b) Minor

i) It would be appreciated if the authors describe this definition as CT slices that include the oral cavity rather than the oral cavity on page 2, lines 57-59.

ii) On page 1, line 36, "DMRs" is thought to be "MDRs".

iii) On page 3, line 95, as "DMAs" is an abbreviation, it needs to be spelled out.

iv) Table 1: The results for “Oral cavity, range (median)”, and “MAs, range (median)” results are misaligned.

v) Table 2: In severities of MAs, the second "mild (< 25%)" is thought to be  "moderate (25%-75%)".

vi) Table 2: In severities of MAs, the P is thought to be italics.

vii) On page 5, line 156, the comma before Moreover is thought to be a period.

Author Response

Thank you for your time and efforts in helping us improve our manuscript and your insightful comments, which were critical in improving the scientific value of our manuscript.

i) It would be appreciated if the authors describe this definition as CT slices that include the oral cavity rather than the oral cavity on page 2, lines 57-59.

We revised this sentence to “In this study, CT slices that include the oral cavity was defined as the area the between the root apex of the maxillary and mandibular canines”.

ii) On page 1, line 36, "DMRs" is thought to be "MDRs".

Thank you very much for your pointing out.

We changed "DMRs" to "MDRs".

iii) On page 3, line 95, as "DMAs" is an abbreviation, it needs to be spelled out.

Thank you very much for your pointing out.

We changed "DMSs" to "MDRs".

iv) Table 1: The results for “Oral cavity, range (median)”, and “MAs, range (median)” results are misaligned.

Thank you very much for your pointing out.

We corrected Table 1 of the misaligned.

v) Table 2: In severities of MAs, the second "mild (< 25%)" is thought to be "moderate (25%-75%)".

Thank you very much for your pointing out.

We revised that to "Moderate.

vi) Table 2: In severities of MAs, the P is thought to be italics.

Thank you very much for your pointing out.

We changed the P to "P.”

vii) On page 5, line 156, the comma before Moreover is thought to be a period.

Thank you very much for your pointing out.

We changed the “comma” to " period”.

Thank you very much for your kind adovice.

Reviewer 2 Report

This paper evaluates the effect of dental metal artifacts in CT for head and neck radiotherapy planning. The research methodology (introduction, methods, results, and discussion) has no major scientific issues.

How about adding one thing: the following

LINE 59-60

”MAs in the oral cavity were classified into three severities by the percentage of CT 59 images containing MAs: mild (< 25%), moderate (25%-75%), and severe (> 75%)”

It would be easier to understand if you could describe the calculation method in more detail.

Thank you in advance.

Author Response

Thank you for your time and efforts in helping us improve our manuscript and your insightful comments, which were critical in improving the scientific value of our manuscript.

i) MAs in the oral cavity were classified into three severities by the percentage of CT 59 images containing MAs: mild (< 25%), moderate (25%-75%), and severe (> 75%).

It would be easier to understand if you could describe the calculation method in more detail.

Thank you very much for your pointing out.

We added the calculation formula to understand easier.

Please check the revised manuscript and short cover letter.
